# Compression Stockings Improve Lower Legs Symptom in Patients with Pulmonary Artery Hypertension Treated by Pulmonary Vasodilators—A Pilot Study

**DOI:** 10.3390/jcm12072484

**Published:** 2023-03-24

**Authors:** Naoko Nakashima, Nobuhiro Tahara, Yoichi Sugiyama, Munehisa Bekki, Shoko Maeda-Ogata, Akihiro Honda, Chidu Goto, Setsuko Tanaka, Yoshihiro Fukumoto

**Affiliations:** 1Nursing Department, Kurume University Hospital, Kurume 830-0011, Japan; 2Division of Cardiovascular Medicine, Department of Internal Medicine, Kurume University School of Medicine, Kurume 830-0011, Japan

**Keywords:** pulmonary arterial hypertension, lower legs symptom, compression stockings

## Abstract

Pulmonary vasodilators have improved pulmonary arterial hypertension (PAH) symptoms and prognosis; however, the drugs cause some side effects, including lower legs pain, which impair quality of life (QOL). The present study examined if compression stockings improved lower extremity symptoms and QOL caused by pulmonary vasodilators in PAH patients. We retrospectively enrolled consecutively ten patients with PAH treated by pulmonary vasodilators, who were regularly followed in Kurume University Hospital from January 2022 to June 2022. Oral questionnaire surveys, the Numeric Rating Scale for Pain (NRS) and the Pain Disability Assessment Scale (PDAS), were conducted regarding lower extremity symptoms before wearing elastic stockings and one month later, to evaluate how the lower extremity symptoms affected daily life. All ten patients were female, with a mean age of 50.2 ± 11.5 years, out of whom intravenous prostacyclin analogue (PGI_2_) was administered in five patients. In no intravenous PGI_2_ group, NRS score was significantly improved from 4.6 ± 2.3 to 2.8 ± 1.2 (*p* = 0.037), while from 9.4 ± 1.2 to 5.4 ± 1.6 (*p* = 0.002) in intravenous PGI_2_ group. PDAS score was also significantly improved [no intravenous PGI_2_ group; 18.0 (15.0–24.0) to 15.0 (10.0–19.0), intravenous PGI_2_ group; 25.0 (17.0–37.0) to 17.0 (5.0–27.0)]. Lower extremity symptoms in patients using pulmonary vasodilators were improved by wearing compression stockings.

## 1. Introduction

Pulmonary arterial hypertension (PAH) is a progressive disease caused by small pulmonary artery obstruction, in which increased pulmonary vascular resistance leads to progressive elevation of pulmonary artery pressure, and right heart failure to death [1]. Because there was no effective therapy in PAH until the mid 1990s, the prognosis was very poor with 1-, 2-, and 3-year survival rates of 77.4%, 51.6%, and 40.6%, respectively [1,2,3]. Since pulmonary vasodilators have been successively developed after mid-1990s, PAH symptoms, exercise tolerance, pulmonary hemodynamics, and prognosis have been significantly improved [1,3,4,5,6].

Pharmacotherapy for PAH has been developed acting on the prostacyclin-cAMP (cyclic adenosine monophosphate), nitric oxide (NO)-cGMP (cyclic guanosine monophosphate), and endothelin pathways. These pulmonary vasodilators are given by oral administration, inhalation, and continuous subcutaneous or intravenous infusion, and the doses are adjusted by escalation to determine the optimal dose for each patient, in consideration of health-related quality of life (QOL) [1]. In the follow-up assessments of PAH, patients are advised to self-monitor for adverse drug events (ADEs) [7]. It is important for patients to appropriately manage ADEs. For example, headache and pain in extremities are often observed in the treatment of PAH, [7] which affect QOL of patients. Especially, lower extremity symptoms seriously affect daily life such as work and housework. The precise mechanisms occurring in lower extremity symptoms are still unclear, and the care for these symptoms have not been developed. Lower extremity symptoms include various clinical features such as sole pain, swelling, dullness, and knee or heel pain. However, many patients have no choice but to endure their lower extremity symptoms.

In our hospital, we recommend the patients with lower extremity symptoms to wear elastic stockings, because a patient, who wore compression stockings, had better lower limbs symptoms. Compression stockings are usually used for patients with varicose veins and lymphedema, and are designed so that the compression pressure is the strongest at the ankle, decreasing toward the center. In compressed lower extremities, pressure improves venous return, heals ulcers, and alleviates symptoms. Therefore, the present study examined the hypothesis if compression stockings improve lower extremity symptoms and QOL caused by pulmonary vasodilators in PAH patients.

## 2. Materials and Methods

### 2.1. Study Design

This study was a single center, retrospective observational study to examine whether compression stockings were effective to reduce lower limbs’ symptoms caused by pulmonary vasodilators. The present study was approved by the institutional review board at Kurume University (21296) and conducted in accordance with the ethical principles of the Declaration of Helsinki. All patients provided written informed consent.

### 2.2. Study Population and Patient Selection

We enrolled consecutive patients aged 20 years or older, who meet the inclusion criteria. Inclusion criteria were as follows: (1) age of ≥20 years, (2) patients with PAH treated by pulmonary vasodilators, who were regularly followed in Cardiovascular Medicine, Kurume University Hospital from January 2022 to June 2022, and (3) those who provided written consent for study participation. Exclusion criteria were as follows: (1) patients with a history of arteriosclerosis obliterans, and (2) patients with varicose vein, lymphedema, and/or skin problems on the lower extremities.

### 2.3. Compression Stockings

For elastic stockings, we measured the lower limbs size of the subjects and selected below-the-knee stockings that fit the size from several companies. Compression pressure was set at 20–29 mmHg, which was commonly used. Before wearing the compression stockings, the lower limbs were observed to check for skin troubles, and the stockings were put on under the direction of the attending cardiologists.

### 2.4. Data Collection

We obtained demographic and clinical information from the electronic medical records of Kurume University Hospital. The patients’ background data including age, sex, height, body weight, Nice clinical classification of PAH, WHO functional classification, blood pressure (BP), and pulse rate, etiology of PAH, 6-min walk test, echocardiography, and right heart catheterization data were collected.

### 2.5. Lower Limbs Symptom Evaluation

Oral questionnaire surveys were conducted regarding lower extremity symptoms, in which NRS and PDAS were used before wearing elastic stockings and one month later, to evaluate how the lower extremity symptoms affected daily life.

#### 2.5.1. Numeric Rating Scale for Pain (NRS)

The pain NRS is an 11-point scale from 0 to 10, which is a unidimensional measure of lower extremities pain intensity [8,9].

#### 2.5.2. Pain Disability Assessment Scale (PDAS)

PDAS was used to assess pain-related disability, on a self-reporting scale from 0 to 3 (0: pain never interfered with these activities; 3: pain completely interfered with these activities), which consists of 20 items; (1) household chores such as vacuuming and gardening, (2) slow running, (3) bending down to pick up something on the floor, (4) going out for shopping, (5) going up and down stairs, (6) visiting friends, (7) taking a bus or train, (8) going out to a restaurant or cafe, (9) carrying heavy stuffs, (10) cooking, washing dishes, (11) bending and stretching waist, (12) reaching out and picking up a heavy object (e.g., sugar bag) from the shelf, (13) washing or wiping body, (14) sitting down and standing up in rest room, (15) getting into bed, getting out of bed, (16) opening and closing car doors, (17) standing still, (18) walking on level ground, (19) hobby activities, and (20) shampoo [10,11].

#### 2.5.3. Specific Comments before and after Compression Stockings

We asked their specific feeling about lower extremity symptoms and location, situations and frequency of lower extremity symptoms, disturbance in life, and other changes before and after compression stockings.

### 2.6. Endpoints

The primary endpoint was the improvement of lower limbs symptoms. The secondary endpoints included specific changes in lower limbs symptoms.

### 2.7. Statistical Analysis

Continuous variables were presented as means ± standard deviation (SD) or median (IQR; interquartile range), as appropriate. They were compared by the Welch’s t-test. Categorical baseline variables were presented as numbers (percentage) and compared by the Fisher’s exact test. All *p* values < 0.05 were considered statistically significant. All analyses were performed with the SPSS system (SPSS Inc., Chicago, IL, USA).

## 3. Results

### 3.1. Patient Characteristics

Ten patients were enrolled in the present study. All 10 patients were female, with a mean age of 50.2 ± 11.5 years. There were 6 idiopathic, 2 hereditary PAH, and 2 PAH associated with connective tissue disease (no scleroderma), who were treated by endothelin receptor antagonist (ERA) in 9, phosphadiesterase-5 (PDE-5) inhibitor in 5, soluble guanylate cyclase (sGC) stimulator in 4, prostacyclin (IP) receptor agonist in 6, oral prostacyclin in 2, and intravenous prostacyclin analogue (PGI_2_) in 5 (Table 1).

### 3.2. Effects of Compression Stockings

We divided the patients into two groups; with and without intravenous PGI_2_ (*n* = 5 each). There were no significant differences other than age in comparing baseline patients’ characteristics, including echocardiography and right heart catheterization data, between with and without intravenous PGI_2_ groups before compression stockings started (Table 2).

In no intravenous PGI_2_ group, all five patients noted fatigue in the lower legs. Three cases occurred while standing and two while walking. Three cases had symptoms every day and two had occasionally (Table 3). In intravenous PGI_2_ group, four patients had pain in the lower extremities, and one had fatigue. The lower extremity symptoms occurred on the femoral, lower leg, and foot. Lower extremity symptoms appeared in four cases when standing. All patients had symptoms every day (Table 3).

Before compression stockings, the NRS score showed 10 (worst score) in four patients in intravenous PGI_2_ group. Compression stockings significantly improved NRS score in both groups, where NRS score was significantly improved from 4.6 ± 2.3 to 2.8 ± 1.2 (*p* = 0.037) in no intravenous PGI_2_ group and from 9.4 ± 1.2 to 5.4 ± 1.6 (*p* = 0.002) in intravenous PGI_2_ group (Figure 1).

The PDAS score also showed similar results (Figure 2). In both groups, PDAS score was significantly improved [no intravenous PGI_2_ group; 18.0 (15.0–24.0) to 15.0 (10.0–19.0), *p* = 0.018, intravenous PGI_2_ group; 25.0 (17.0–37.0) to 17.0 (5.0–27.0), *p* = 0.046]. In terms of the specific comments of the PDAS, all five subjects in no intravenous PGI_2_ group showed difficulty in performing “chores such as vacuuming and gardening”, “carrying heavy stuffs”, and “walking on level ground”, and four patients improved in the difficulty of “walking on a level ground” after compression stockings. In the intravenous PGI_2_ group, all five patients felt difficulty in the following items: “going out for shopping”, “going up and down stairs”, “carrying heavy stuffs”, and “walking on level ground”. After compression stockings, all five patients improved in “going out for shopping” and “walking on level ground”, and four of them improved in “going up and down stairs”.

### 3.3. Specific Symptoms before and after Compression Stockings

Before compression stockings, their complaints were “It is hard to stand up. Shopping is pain.”, “I can’t play with my kids after I pick up from kindergarten. It is hard to prepare dinner.”, and “I work standing. In 30 min, my legs start to hurt, and I cry.” For patient-specific symptom relief, we have recommended self-care such as resting, sitting down, resting with legs up, or massage. After the patients started compression stockings, they have changed to “At the moment I put them on, I feel my legs light, completely different! Edema and leg cramp have reduced.”, “I feel less pain when working. I can come home after shopping.”, and “I can walk!”, suggesting reduction of pain when shopping or working, as well as leg edema. However, there was no change in knee or heel pain. In addition, there were some complaints from patients that it was some difficult to wear compression stockings and also that it was stuffy. During the observational period, no varicose vein, lymphedema, or skin problems were observed in the lower extremities in the enrolled patients.

## 4. Discussion

This is the first report to demonstrate the favorable effects of compression stockings on lower limbs symptoms in PAH patients. Pulmonary vasodilators for PAH treatment often cause side effects in their lower extremities, but the patients have always given them up. We have usually recommended only self-care such as resting, sitting down, or massage. Many PAH patients have had difficulties in shopping, housework, and working out, due to lower extremity symptoms. Now, we are able to recommend the compression stockings to relieve the lower limbs symptom caused by pulmonary vasodilators.

### 4.1. Lower Limbs Symptoms in PAH Patients

Although the prognosis of PAH has been remarkably improved as therapeutic options have developed more and more for these two decades, [6] ADEs caused by pulmonary vasodilators commonly occurred in PAH patients, which might be dose-dependent and dose-related [12]. However, ADEs in early phase of PAH treatment was not associated with the improved mortality [12]. Therefore, ADEs, which impair the patients’ QOL, are not welcome and should be avoided. ADEs caused by pulmonary vasodilators have been reported as headache, gastrointestinal symptoms, jaw pain, and extremity pain [7,13,14]. Especially, pain in lower limbs brings big problems in daily lives, which should be relieved.

### 4.2. Compression Stockings

Compression stockings are used for lower extremity varicose veins, lymphedema, and other venous and lymphatic diseases, which are effective in relieving symptoms such as lower extremity swelling and dullness [15]. As the present study indicated, pulmonary vasodilators impair the activities of daily lives in many patients with PAH due to lower extremity symptoms, such as walking, shopping, childcare, housework, and working. It has been shown that compression stockings are effective in reducing pain and symptoms in patients suffering from chronic venous insufficiency, [16] because the compression therapy improves venous pumping by reducing the venous cross sectional diameters, the contraction of the calf muscles to press the veins of the lower extremities, and squeezing out the blood flow, all of which improve venous return [17]. Compression therapy may also work on leg edema improvement, which is caused by right heart failure in PAH patients [18]. Congestion in the lower extremities causes various subjective symptoms such as dullness, hot flushes, pain, swelling, itching, and cramps during sleep, which can be improved by compression stockings. In particular, the intravenous PGI_2_ group had worse symptoms and difficulties, in whom wearing elastic stockings were also able to alleviate the symptoms and to reduce the difficulties in daily lives.

We have a patient experience first that wearing elastic stockings has reduced the pain in her lower limbs and made her easier to work, and then confirmed the effects in ten patients. The adverse effects of pulmonary vasodilators have not been focused much; however, now that the prognosis of PAH has been improved, [4,5,6] we have to look at the ADEs of pulmonary vasodilators.

There are some contraindications in compression stockings, which should not be offered to patients with severe congestive cardiac failure, due to a risk that systemic fluid overload might be developed [15]. However, reducing moderate edemas, without shifting larger blood volumes toward the right heart, can be achieved using light compression stockings [15]. Further in patients with critical limb ischemia, with systolic ankle pressure below 70 mmHg or following arterial bypass grafting, compression stockings are contraindicated as there is a risk of ischemia or local skin necrosis, especially if the limb is elevated [15]. Other damages, which may be caused by stockings, include advanced peripheral neuropathy, fragile tissue paper skin over the bony prominences, dermatitis, and allergic reactions to the fabric [15]. In such cases, compression stockings should be used and customized to the individual limb measurements.

### 4.3. Limitations

The present study has several limitations. First, the present study had a small number of enrolled subjects, and the effect of elastic stockings could not be sufficiently clarified. Second, this was a single-arm, open label, and retrospective study, with a potential bias. Third, we need to prospectively clarify the beneficial effects of compression stockings in a larger population. We are planning to perform the multi-center, prospective study, considering careful indication of compression stockings [15].

## 5. Conclusions

Lower limbs symptoms were strong in PAH patients treated by pulmonary vasodilators. Compression stockings attenuated their lower extremities symptoms and improved their quality of lives.

## Figures and Tables

**Figure 1 jcm-12-02484-f001:**
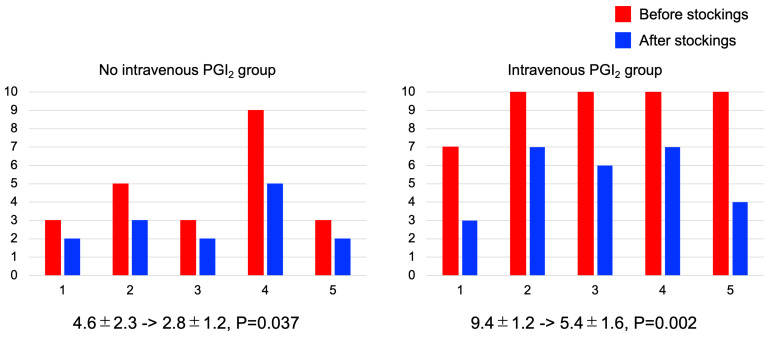
Effects of compression stockings evaluated by Numeric Rating Scale for Pain (NRS). In intravenous prostacyclin analogue (PGI_2_) group, four of five patients had score 10 (worst) before stockings. Wearing compression stockings significantly improved NRS score in both groups.

**Figure 2 jcm-12-02484-f002:**
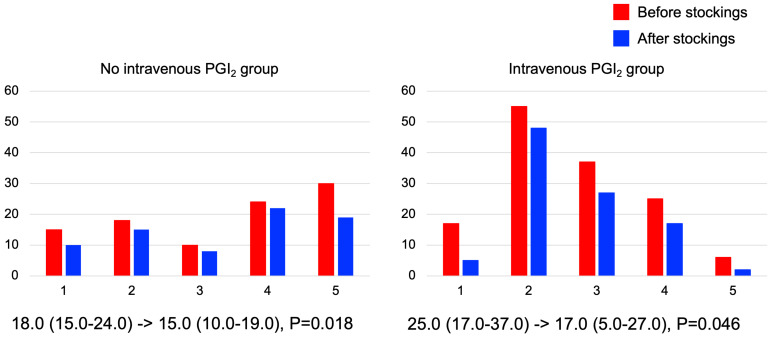
Effects of compression stockings evaluated by Pain Disability Assessment Scale (PDAS). Wearing compression stockings significantly improved PDAS score in both groups.

**Table 1 jcm-12-02484-t001:** Clinical characteristics.

N	10
Female	10
Age, years old	50.2 ± 11.5
Height, cm	153.9 ± 4.5
Weight, kg	52.6 ± 15.2
SBP, mmHg	97.5 ± 10.7
DBP, mmHg	60.5 ± 9.8
Pulse rate, /min	75.7 ± 14.4
Diagnosis	
IPAH	6
HPAH	2
CTD-PAH	2
WHO functional class	
I/II/III, *n*	2/5/3
Medication	
ERA	9
PDE-5 inhibitor	5
sGC stimulator	4
IP receptor agonist	6
Oral prostacyclin	2
Intravenous PGI_2_	5

Data are mean ± SD, or *n*. SBP; systolic blood pressure, DBP; diastolic blood pressure, IPAH; idiopathic pulmonary arterial hypertension, HPAH; heritable pulmonary arterial hypertension, CTD-PAH; pulmonary arterial hypertension associated with connective tissue disease, ERA; endothelin receptor antagonist, PDE-5; phosphadiesterase-5, sGC; soluble guanylate cyclase, IP receptor; prostacyclin receptor, PGI_2_; prostacyclin analogue.

**Table 2 jcm-12-02484-t002:** Comparison between with and without intravenous prostacyclin.

	Overall	No Intravenous PGI_2_	Intravenous PGI_2_	*p* Value
	*n* = 10	*n* = 5	*n* = 5	
Age, years old	50.2 ± 10.9	58.0 ± 6.0	42.4 ± 9.0	0.020
WHO functional class				
I/II/III	2/5/3	0/3 2	2/2/1	0.282
6 min walk, m	485 ± 96	440 ± 105	531 ± 58	0.165
Echocardiography				
TAPSE, mm	26.6 ± 8.6	31.7 ± 9.5	21.6 ± 2.2	0.072
FAC, %	36.7 ± 7.9	33.6 ± 7.1	39.8 ± 7.5	0.265
TRPG, mmHg	38.9 ± 18.3	31.0 ± 5.8	46.8 ± 22.5	0.212
Right heart catheterization				
PAP, mmHg	30.7 ± 11.7	27.4 ± 8.3	34.0 ± 13.5	0.430
RAP, mmHg	5.0 ± 2.8	4.2 ± 3.5	5.8 ± 1.2	0.416
CI, L/min/m^2^	3.78 ± 1.55	3.48 ± 1.73	4.09 ± 1.28	0.586
PVR, dynes·sec·cm^−5^	283 (120–572)	278 (105–792)	354 (115–762)	0.865

Data are mean ± SD, *n*, or median (interquartile range). *p* value; compared between with and without intravenous PGI_2_ groups. IPAH; idiopathic pulmonary arterial hypertension, HPAH; heritable pulmonary arterial hypertension, CTD-PAH; pulmonary arterial hypertension associated with connective tissue disease, ERA; endothelin receptor antagonist, PDE-5; phosphadiesterase-5, sGC; soluble guanylate cyclase, IP receptor; prostacyclin receptor, PGI_2_; prostacyclin analogue.

**Table 3 jcm-12-02484-t003:** Patients’ clinical characteristics and symptoms.

	Sex	Onset, Year	Diagnosis	WHO-FC	Lower Extremities Symptom	Appearance	Frequency
**No Intravenous PGI_2_**							
1	Female	2000	IPAH	II	fatigue, lower-leg	walking	sometimes
2	Female	2017	CTD-PAH	III	fatigue, lower-leg	standing	everyday
3	Female	2017	IPAH	II	fatigue, lower-leg	walking	everyday
4	Female	2009	CTD-PAH	III	fatigue, lower-leg	standing	everyday
5	Female	2017	IPAH	II	fatigue, lower-leg	standing	sometimes
**Intravenous PGI_2_**							
1	Female	2016	HPAH	II	fatigue, femoral and lower-leg	standing	everyday
2	Female	2018	IPAH	I	pain, femoral and foot	walking	everyday
3	Female	2000	HPAH	II	pain, foot	standing	everyday
4	Female	2009	IPAH	III	pain, lower-leg	standing	everyday
5	Female	2016	IPAH	II	pain, lower-leg and foot	standing	everyday

IPAH; idiopathic pulmonary arterial hypertension, HPAH; heritable pulmonary arterial hypertension, CTD-PAH; pulmonary arterial hypertension associated with connective tissue disease, FC; functional class.

## Data Availability

The data presented in this study are available on request from the corresponding author. The data are not publicly available due to privacy restrictions.

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
