# Peer review of "Compression Stockings Improve Lower Legs Symptom in Patients with Pulmonary Artery Hypertension Treated by Pulmonary Vasodilators—A Pilot Study"

_jcm, 2023, doi:10.3390/jcm12072484_

Round 1
Reviewer 1 Report
This is a nice pilot study showing the benefit of compression in this group.
However it is not clear from the text if any of the patients had additional venous or lymphatic disease. This is very frequent in this age group.
This should be commented in the text and in a future study those patients should be excluded or for a subgroup.
Author Response
Responses to the Reviewer 1
Manuscript: jcm-2261533_R1
Authors: Naoko Nakashima, et al.
Title: Compression stockings improve lower legs symptom in patients with pulmonary artery hypertension treated by pulmonary vasodilators; a pilot study
We thank the Reviewer for the valuable comments. In line with the comment, we have revised our manuscript. Our detailed responses will follow the Reviewer’s comments. Our point-to-point responses are shown in the text in red to facilitate the review process.
Reviewer 1 comments:
This is a nice pilot study showing the benefit of compression in this group.
However, it is not clear from the text if any of the patients had additional venous or lymphatic disease. This is very frequent in this age group.
This should be commented in the text and in a future study those patients should be excluded or for a subgroup.
[Response]
Thank you for your valuable comment. We fully agree with the Reviewer that this age group has often venous and/or lymphatic diseases. When we use compression stockings, we carefully check if the patients have varicose vein, lymphedema, and/or other skin symptoms. In the enrolled patients, there were no patients with these diseases. According to the Reviewer’s comment, we have revised the paper as described below.
Lines 76-78: Exclusion criteria were as follows: 1) patients with a history of arteriosclerosis obliterans, and 2) patients with varicose vein, lymphedema, and/or skin problems on the lower extremities.
Lines 196-198: During the observational period, no varicose vein, lymphedema, or skin problems was observed in the lower extremities in the enrolled patients.
Lines 239-249: There are some contraindications in compression stockings, which should not be offered to patients with severe congestive cardiac failure, due to a risk that systemic fluid overload might be developed.[15] However, reducing moderate edemas, without shifting larger blood volumes towards the right heart, can be achieved using light compression stockings.[15] Further in patients with critical limb ischemia, with systolic ankle pressure below 70 mmHg or following arterial bypass grafting, compression stockings are contraindicated as there is a risk of ischemia or local skin necrosis, especially if the limb is elevated.[15] Other damages, which may be caused by stockings, include advanced peripheral neuropathy, fragile tissue paper skin over the bony prominences, dermatitis and allergic reactions to the fabric.[15] In such cases, compression stockings should be used and customized to the individual limb measurements.
Lines 255-256: We are planning to perform the multi-center, prospective study, considering careful indication of compression stockings.[15]
Finally, we again would like to thank the Reviewer for the valuable comments on our work. We sincerely hope that our revised manuscript may again be considered for publication in the Journal.

Reviewer 2 Report
This is a small and simple study, in which PAH patients are given compression stockings to alleviate side effects of a common side effect of prostacyclin, lower leg pain. Although compression stockings are mentioned in some other papers as a way of dealing with this side effect, to my knowledge (and after a brief search, as far as I can tell from pubmed), this is the first study of any kind to assess effectiveness.
Although the study is quite small, the paired design gives it sufficient power to draw at least tentative conclusions. I've looked at all of the figures, methods, and tables carefully, and I see no changes that need to be made. All of the drawbacks to the study are clearly listed by the authors.
Author Response
Responses to the Reviewer 2
Manuscript: jcm-2261533_R1
Authors: Naoko Nakashima, et al.
Title: Compression stockings improve lower legs symptom in patients with pulmonary artery hypertension treated by pulmonary vasodilators; a pilot study
Reviewer 2 comments:
This is a small and simple study, in which PAH patients are given compression stockings to alleviate side effects of a common side effect of prostacyclin, lower leg pain. Although compression stockings are mentioned in some other papers as a way of dealing with this side effect, to my knowledge (and after a brief search, as far as I can tell from pubmed), this is the first study of any kind to assess effectiveness.
Although the study is quite small, the paired design gives it sufficient power to draw at least tentative conclusions. I've looked at all of the figures, methods, and tables carefully, and I see no changes that need to be made. All of the drawbacks to the study are clearly listed by the authors.
[Response]
Thank you for your favorable comment.
We sincerely hope that our manuscript may again be considered for publication in the Journal.
